



# On the treatment of discordant detrital zircon U–Pb data

Pieter Vermeesch[1]

[1]Department of Earth Sciences, University College London, Gower Street, London WC1E 6BT

**Correspondence:** Pieter Vermeesch (p.vermeesch@ucl.ac.uk)

**Abstract.** Detrital zircon U–Pb geochronology is a staple of sedimentary provenance analysis and crustal evolution studies. Constructing detrital age spectra is straightforward for concordant $^{206}$Pb/$^{238}$U- and $^{207}$Pb/$^{206}$Pb-compositions. But unfortunately, many detrital U–Pb datasets contain a significant proportion of discordant analyses. This paper investigates two decisions that must be made when analysing such discordant U–Pb data.

First, the analyst must choose whether to use the $^{206}$Pb/$^{238}$U- or the $^{207}$Pb/$^{206}$Pb-date. The $^{206}$Pb/$^{238}$U-method is more precise for young samples, whereas the $^{207}$Pb/$^{206}$Pb-method is better suited for old samples. However there is no agreement which 'cutoff' should be used to switch between the two. This subjective decision can be avoided by using single grain concordia ages. These represent a kind of weighted mean between the $^{206}$Pb/$^{238}$U- and $^{207}$Pb/$^{206}$Pb-methods, which offers better precision than either of the latter two methods.

A second subjective decision is how to define the discordance cutoff between 'good' and 'bad' data. Discordance is usually defined as (1) the relative age difference between the $^{206}$Pb/$^{238}$U and $^{207}$Pb/$^{206}$Pb dates. However, this paper shows that several other definitions are possible as well, including (2) the absolute age difference; (3) the common-Pb fraction according to the Stacey-Kramers mantle evolution model; (4) the p-value of concordance; (5) the perpendicular logratio (or 'Aitchison') distance to the concordia line; and (6) the logratio distance to the maximum likelihood composition on the concordia line.

Applying these six discordance filters to a 10,000-grain dataset of detrital zircon U–Pb compositions reveals that: (i) the relative age discordance filter tends to suppress the young age components in U–Pb age spectra, whilst inflating the older age components; (ii) the Stacey-Kramers discordance filter is more likely to reject old grains and less likely to reject young ones; (iii) the p-value based discordance filter has the undesirable effect of biasing the results towards the least precise measurements; (iv) the logratio-based discordance filters are most strict for Proterozoic grains, and more lenient for Phanerozoic and Archaean age components; (v) of all the methods, the logratio distance to the concordia composition produces the best results, in the sense that it yields nearly identical $^{206}$Pb/$^{238}$U and $^{207}$Pb/$^{206}$Pb age spectra without introducing an age bias. All the methods presented in this paper have been implemented in the `IsoplotR` toolbox for geochronology.

## 1   Introduction

The U–Pb method consists of two paired decay systems, in which two isotopes of the same radioactive parent ($^{238}$U and $^{235}$U) decay to two isotopes of the same radiogenic daughter ($^{206}$Pb and $^{207}$Pb, respectively). This paired decay system provides a powerful internal consistency check for the method, which is absent from other chronometers. By 'double dating' samples





with the $^{206}$Pb/$^{238}$U and $^{207}$Pb/$^{235}$U methods (or, equivalently, the $^{206}$Pb/$^{238}$U and $^{207}$Pb/$^{206}$Pb-methods) it is possible to verify whether the isotopic system is free of primary or secondary disturbances. The most reliable age constraints are obtained from samples whose $^{206}$Pb/$^{238}$U, $^{207}$Pb/$^{235}$U and $^{207}$Pb/$^{206}$Pb ages are statistically indistinguishable from each other. U–Pb

compositions that fulfil this requirements are 'concordant'. Those that violate it are 'discordant'.

Discordance can be caused by a number of mechanisms, including: (a) the presence of non-radiogenic ('common') lead; (b) initial disequilibrium between the short-lived nuclides of the $^{238}$U–$^{206}$Pb and $^{235}$U–$^{207}$Pb decay chains; (c) partial loss of radiogenic lead during high grade metamorphism; and (d) mixing of different age domains during micro-analysis (Schoene, 2014). These complicating effects can often be diagnosed and remediated when multiple cogenetic crystals are available from

the same sample. If the aliquots form an isochron (or 'discordia') line in $^{238}$U-$^{204,6,7}$Pb space, then this line can be used to recover robust chronologies from discordant data (Ludwig, 1998).

Unfortunately, this procedure is rarely or never possible for detrital samples, in which crystals of datable minerals are not guaranteed to be cogenetic. Without a universal mechanism to identify the cause of U–Pb discordance and remove its effects, detrital geochronologists have no choice but to accept some discordant analyses and somehow incorporate them into their

age spectra. There exists a lack of consensus among the detrital zircon geochronology community on how to do this. Two outstanding questions are:

1. Which age estimate to use? It is widely recognised that $^{206}$Pb/$^{238}$U age estimates offer the optimal accuracy and precision at the young end of the age spectrum, whereas the $^{207}$Pb/$^{206}$Pb method is better suited for older samples. However the cutoff between the two clocks varies between studies, with values ranging from 800 Ma to 1.5 Ga (Gehrels, 2011;

Spencer et al., 2016).

2. How to quantify discordance? Most studies define discordance as the relative age difference between the $^{206}$Pb/$^{238}$U and $^{207}$Pb/$^{206}$Pb ages, but some advocate the use of statistical hypothesis tests and p-values to quantify discordance (Spencer et al., 2016). And even when a discordance definition has been agreed upon, there are many ways to choose the discordance cutoff. For example, the relative age discordance threshold may vary between 10% and 30% (Gehrels,

50    2011).

This paper addresses both of these issues. Section 2 advocates the use of single-grain concordia ages (Ludwig, 1998) as a way to avoid the arbitrary cutoff between the $^{206}$Pb/$^{238}$U and $^{207}$Pb/$^{206}$Pb methods. Although previous workers have argued for the use of single-grain concordia ages before (see Zimmermann et al., 2018, for a recent example), this study uses a semi-analytical model, rather than purely empirical arguments, to demonstrate the superior precision of this hybrid chronometer.

Section 3 compares and contrasts existing discordance filters based on age disparity and p-values. It shows that the relative age definition strongly favours older samples over young ones, and that the p-value definition, which has gained popularity in recent years, hurts both the accuracy and precision of detrital geochronology. The age disparity and p-value definitions are heuristic by nature and are not based on firm statistical or geological arguments. Although they are essentially the only two definitions of discordance that are used in detrital geochronology today, they are by no means the only two possible options.





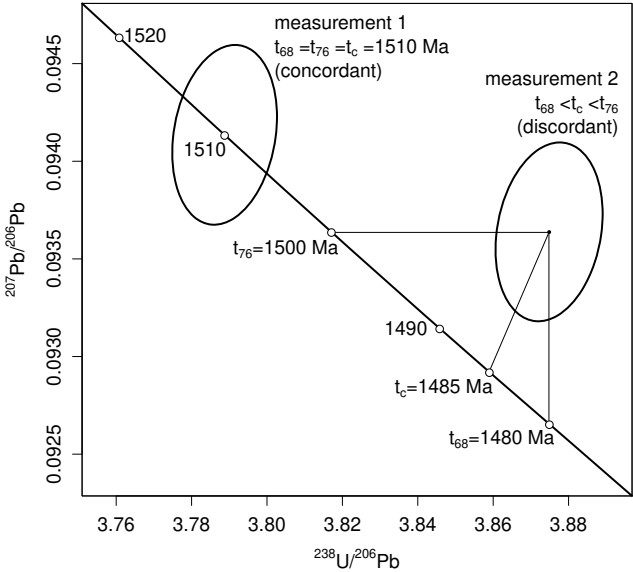

**Figure 1.** Illustrative Tera-Wasserburg concordia diagram with a concordant and discordant measurement. $t_{68}$ marks the $^{206}Pb/^{238}U$ age, $t_{76}$ the $^{207}Pb/^{206}Pb$ age, and $t_c$ the concordia age. Measurement 1 is concordant because its estimates for $t_{68}$, $t_{76}$ and $t_c$ are identical. Measurement 2 is discordant because the three estimates disagree. The concordia age is the most likely age given the analytical uncertainties. It falls between the other two age estimates, and offers the best analytical precision of the three.

Section 4 addresses the inherent biases of the existing discordance definitions by proposing three new definitions, which are based directly on U–Pb compositions rather than on the ages calculated therefrom. The first new definition assumes that the discordance is caused by the presence of common lead. The other two new definitions treat U–Pb discordance as a compositional data problem (*sensu* Aitchison, 1986). Isotope ratios are strictly positive quantities and log-contrasts are the 'natural' way to quantify 'distances' between them. Section 4 introduces two logratio definitions of discordance, ignoring and accounting for 65 analytical uncertainty, respectively.

Although the new definitions are arguably more attractive than the old ones from a theoretical point of view, this does not guarantee that they produce more sensible results. To test their performance on real data, Section 5 applies all six discordance filters to a compilation of detrital zircon U–Pb data. Although the true age distribution of this dataset is unknowable, the results suggest that the logratio based discordance filters produce the most accurate, and most easily interpretable results. The p-value 70 definition fares the worst and is not recommended.

## 2 Which age to choose?

The U–Pb method is based on three separate chronometers: $^{206}Pb/^{238}U$, $^{207}Pb/^{235}U$ and $^{207}Pb/^{206}Pb$. The half-life of $^{235}U$ is more than six times shorter than that of $^{238}U$, and $^{235}U$ is more than 100 times less abundant than $^{238}U$. For these two reasons,





little $^{207}$Pb has been produced during the last billion years of Earth history compared to $^{206}$Pb. Consequently, the $^{207}$Pb/$^{235}$U
and $^{207}$Pb/$^{206}$Pb methods are less precise than the $^{206}$Pb/$^{238}$U method during the Phanerozoic and Neoproterozoic.

However, during earlier stages of Earth's history, $^{235}$U was significantly more abundant than it is today. The $^{238}$U/$^{235}$U ratio
was ~60 at 1Ga, ~26 at 2Ga, ~11 at 3Ga, and ~5 at 4Ga. Due to the greater abundance of $^{235}$U in this past, and because it
decays much faster than $^{238}$U, the precision of the $^{207}$Pb/$^{235}$U and $^{207}$Pb/$^{206}$Pb clocks exceeds that of the $^{206}$Pb/$^{238}$U method
during the Palaeoproterozoic and Archaean. The gradual shift in sensitivity between the two chronometers is visible in the
slope of a Tera-Wasserburg concordia line, which is steep at old ages (high $^{207}$Pb/$^{206}$Pb gradient w.r.t. time) and shallow at
young ages (low $^{238}$U/$^{206}$Pb gradient w.r.t. time).

Most published detrital zircon U–Pb studies switch from $^{206}$Pb/$^{238}$U to $^{207}$Pb/$^{206}$Pb at some point during the Proterozoic.
Unfortunately there are two problems with such a switch. First, it requires the selection of a discrete discordance cutoff between
the two methods. If this cutoff differs between two studies (which it often does), then this complicates the intercomparison of
their respective age spectra. Second, the sudden switch between the $^{206}$Pb/$^{238}$U and $^{207}$Pb/$^{206}$Pb clocks is often marked by a
discrete step in the age spectrum (Puetz et al., 2018). This step is entirely artificial and obscures any geologically significant
events that might occur around the same time.

Both of these problems can be solved by using 'hybrid' concordia ages instead of 'pure' $^{206}$Pb/$^{238}$U and $^{207}$Pb/$^{206}$Pb ages.
Concordia ages are defined by Ludwig (1998) as the 'most likely' (in a statistical sense) U–Pb age given the isotopic ratio
composition and its analytical uncertainty (Figure 1). Let $r_{75}$ and $r_{68}$ be the measured $^{207}$Pb/$^{235}$U and $^{206}$Pb/$^{238}$U ratios, re-
spectively, and let $\sigma[r_{68}]^2$, $\sigma[r_{76}]^2$, $\sigma[r_{68}, r_{76}]$ be their (co)variances. Then the concordia age $t_c$ is obtained by numerically
minimising the sum of squares $S$:

$$S = \begin{bmatrix} r_{75} - \exp(\lambda_{235}t_c) + 1 \\ r_{68} - \exp(\lambda_{238}t_c) + 1 \end{bmatrix}^T \begin{bmatrix} \sigma[r_{75}]^2 & \sigma[r_{75}, r_{68}] \\ \sigma[r_{75}, r_{68}] & \sigma[r_{68}]^2 \end{bmatrix}^{-1} \begin{bmatrix} r_{75} - \exp(\lambda_{235}t_c) + 1 \\ r_{68} - \exp(\lambda_{238}t_c) + 1 \end{bmatrix} \tag{1}$$

The single grain concordia age combines the chronometric power of the $^{206}$Pb/$^{238}$U and $^{207}$Pb/$^{206}$Pb systems. For young
(<1 Ga) samples, the concordia age is nearly identical to the $^{206}$Pb/$^{238}$U age. For old samples (>2 Ga) it approaches the
$^{207}$Pb/$^{206}$Pb age. For Proterozoic samples, the concordia age gradually shifts from the $^{206}$Pb/$^{238}$U to the $^{207}$Pb/$^{206}$Pb age. Us-
ing concordia ages removes the need for an arbitrary cutoff between the two chronometers. An additional advantage is that
the concordia age offers better precision than the $^{206}$Pb/$^{238}$U and the $^{207}$Pb/$^{206}$Pb chronometer (or the $^{207}$Pb/$^{235}$U for that
matter). Figure 2 quantifies this effect using a semi-analytical mass spectrometry simulation whose algorithm is provided in
Appendix A.

## 3  Discordance filters: old definitions

The most common definition of discordance uses the relative difference between the $^{206}$Pb/$^{238}$U and $^{207}$Pb/$^{206}$Pb age estimate
(Gehrels, 2011):

$$d_r = 1 - t_{68}/t_{76} \tag{2}$$





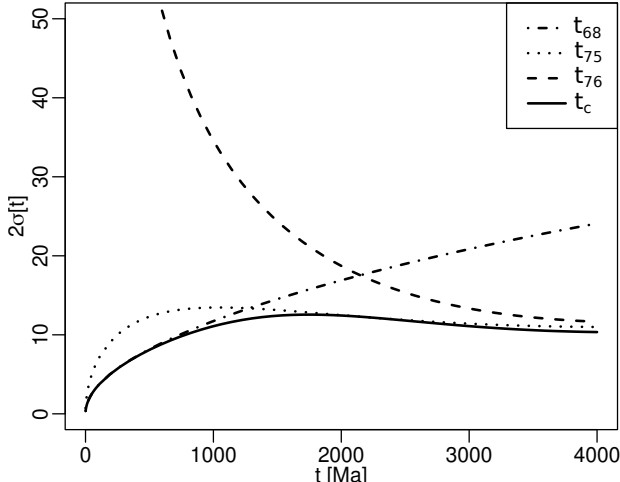

**Figure 2.** Predicted uncertainties of the $^{206}$Pb/$^{238}$U ($t_{68}$), $^{207}$Pb/$^{235}$U ($t_{75}$), $^{207}$Pb/$^{206}$Pb ($t_{76}$) and concordia ($t_c$) ages for a synthetic dataset with a constant uranium concentration. Dwell times and detector sensitivities were chosen so as to yield results that are similar to those obtained from real data. The concordia age (solid line) always offers the best precision. See Appendix A for further details about the calculations behind this figure.

However other definitions are possible as well. For example, one could also define discordance in terms of absolute age differences (Puetz et al., 2018):

$$d_t = t_{76} - t_{68} \tag{3}$$

A third option is to define discordance in terms of U–Pb compositions rather than ages. Spencer et al. (2016) advocate using p-values to assess concordance. In the context of single grain concordia ages, the p-value is the probability that the sum of

squares $S$ (Equation 1) exceeds the observed value under a chi-square distribution with two degrees of freedom:

$$d_p = \text{Prob}\left(s > S | S \sim \chi_2^2\right) \tag{4}$$

Detrital zircon U–Pb data can be filtered by removing all measurements whose discordance values exceed a certain threshold value. Typical cutoff values for $d_r$ are 10–30% (Gehrels, 2011), whereas $d_p$ is generally set to 5% (Spencer et al., 2016). Different discordance criteria produce different U–Pb age spectra. For example, a relative age cutoff will preferentially remove

young grains whereas an absolute age cutoff is comparatively more likely to remove old grains (Figure 3).

The p-value definition affects grains differently depending on their analytical precision (Nemchin and Cawood, 2005). For example, consider a 1.5 Ga zircon that is $d_r = 1\%$ discordant. If this grain were analysed by LA-ICP-MS with an analytical precision of 2%, say, then it would pass the chi-square test and be accepted as being concordant. However, if that same grain were analysed by TIMS with a precision of 0.2%, then the p-value criterion would reject it as being discordant (Figure 4).

It seems fundamentally unfair that an imprecise analytical method would be favoured over a precise one. This is a pertinent



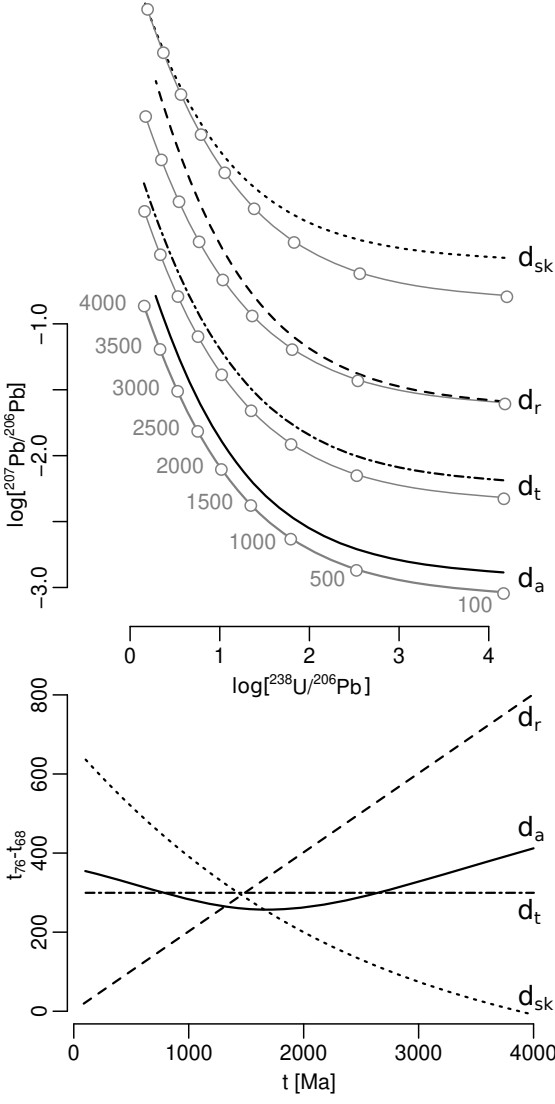

**Figure 3.** Discordance cutoffs for four of the six discordance definitions discussed in Sections 3 and 4. The $d_p$ and $d_c$ criteria are not shown because they depend on the analytical uncertainty of the measurements, which may vary between studies. The top panel shows cutoffs of $d_r = 20\%$ (relative age filter, dashed line), $d_t = 300$ Ma (absolute age filter, dash-dot line), $d_{sk} = 2\%$ (Stacey–Kramers filter, dotted line) and $d_a = 15\%$ (Aitchison distance, solid line) on a Tera-Wasserburg concordia diagram, which is plotted in logarithmic space to provide a more balanced view of the old and young ends of the time scale. The bottom panel presents the $^{207}$Pb/$^{206}$Pb $-$ $^{206}$Pb/$^{238}$U age discordance along this line. This shows that the $d_r$-criterion favours young samples and the $d_{sk}$-criterion old samples, whereas the $d_a$-criterion is most strict for samples of intermediate (i.e., Proterozoic) age.

problem because technical innovations are increasing the precision of all analytical approaches to U–Pb geochronology. As



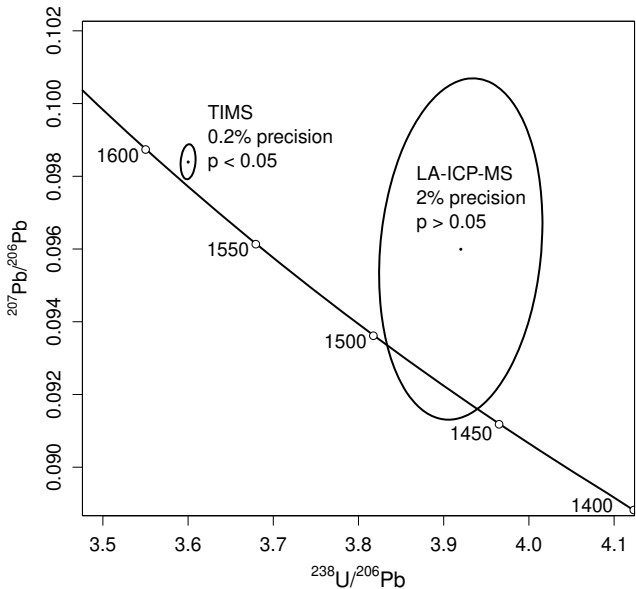

**Figure 4.** Application of the flawed p-value discordance criterion to two synthetic measurements by TIMS (left) and LA-ICP-MS (right). The precise TIMS measurement is labelled as discordant even though it plots closer to the concordia line than the imprecise LA-ICP-MS measurement, which is labelled as concordant.

precision improves, so does the ability to detect ever small degrees of discordance. Using the p-value criterion, there may come a time when no detrital zircon passes this filter.

A final argument against the p-value discordance criterion is that it biases against old U–Pb ages. This is because old zircon

contains more radiogenic Pb than young zircon does. Therefore the analytical precision of the isotopic ratio measurements tends to be better for old grains than it is for young ones. Consequently, the chi-square test has greater power (*sensu* Cohen, 1992) to reject them. In conclusion, p-value based discordance filters are fundamentally flawed. Despite their appeal as 'objective' tools for statistical decision making, formalised hypothesis tests such as chi-square are rarely useful in geology. For the same reason, the widely used MSWD (Mean Square of the Weighted Deviates, McIntyre et al., 1966) statistic (which is just $S/2$ in

this case) should be used with caution. This is because, like p-values, also MSWD cutoffs punish precise datasets in favour of imprecise ones. Note that this caveat also goes against the recommendations of Spencer et al. (2016).

## 4 Discordance filters: new definitions

Section 3 reviewed three existing discordance definitions. This Section will introduce three new ones. None of the definitions discussed thus far encode any information about the geological mechanisms behind the discordance. As explained in Section 1,

common Pb is one of the most likely causes of discordance. Using a mantle evolution model (e.g. Stacey and Kramers, 1975)





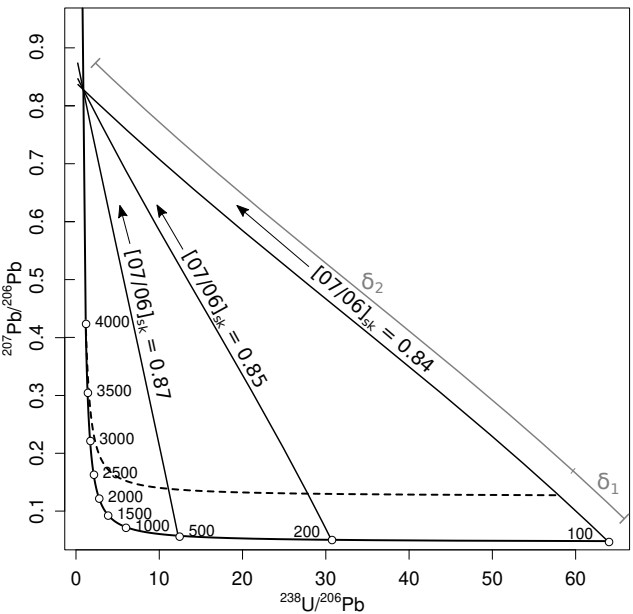

**Figure 5.** Using the Stacey and Kramers (1975) common Pb model as a discordance criterion. This criterion assumes that the discordance is caused by linear mixing (hence, the linear scale of this Tera-Wasserburg plot) between radiogenic Pb (intersections of the mixing lines with concordia) and common Pb (intersection of the mixing lines with the vertical axis). The dashed line marks the 20% ($= \delta_1/[\delta_1 + \delta_2]$) discordance cutoff. This discordance filter, which must be applied *before* making any actual common Pb correction, is more forgiving for young grains than it is for old grains. In this respect, it has the opposite effect of the relative age filter shown in Figure 3.

to approximate the isotopic composition of this common Pb, discordance can be defined as:

$$d_{sk} = 1 - \left[\frac{^{238}U}{^{206}Pb}\right] \Big/ \left[\frac{^{238}U}{^{206}Pb}\right]^* \tag{5}$$

where $\left[^{238}U/^{206}Pb\right]^*$ is the $^{238}U/^{206}Pb$-ratio of the intersection between concordia and a straight line connecting the $^{238}U/^{206}Pb$–$^{207}Pb/^{206}Pb$ measurement to the inferred mantle composition (Figure 5).

The common Pb definition of discordance is more forgiving for young grains than it is for old ones. Importantly, if the discordance is caused by common Pb, then the $^{206}Pb/^{238}U$, $^{207}Pb/^{206}Pb$ and concordia age estimates are all negatively biased with respect to the true age. However this bias can be removed by applying a common-Pb correction *after* the data have been filtered.

Each discordia definition that we have studied thus far is expressed in different units. For the absolute age definition, degrees of discordance are expressed in units of time (ranging from 0 to 4.5 Ga). The relative age definition uses fractions of time (ranging from $-\infty$ to 1). The p-value definition expresses discordance in terms of probability (ranging from 0 to 1). And the Stacey and Kramers (1975) definition uses fractions of ratios (ranging from $-\infty$ to 1). None of these scales is particularly intuitive or natural. They certainly do not match the usual definition of *distance* in the geographical sense of the word.




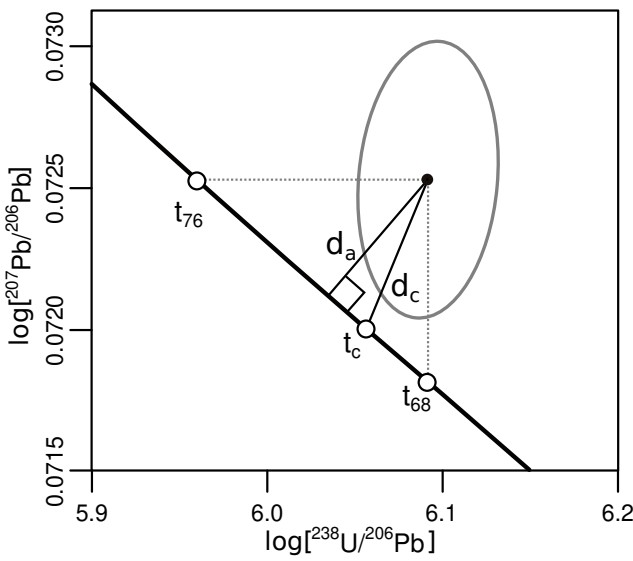

**Figure 6.** Illustration of the two logratio distance definitions of discordance. $d_a$ is the perpendicular Aitchison distance from the measured logratio to the concordia line. $d_c$ is the Aitchison distance measured along a line connecting the measured value and the concordia composition.

To address this issue, it is useful to subject the U–Pb isotopic ratio data to a logarithmic transformation. So instead of
analysing our data on a conventional Tera-Wasserburg concordia diagram, all calculations can be done in $\log(^{207}\text{Pb}/^{206}\text{Pb})$ vs.
$\log(^{238}\text{U}/^{206}\text{Pb})$ space. The advantage of this transformation is that it produces values that are free to range from $-\infty$ to $+\infty$.
Within this infinite dataspace, the Euclidean distance metric can be safely applied.

There exists a vast body of statistical literature detailing the theoretical and practical advantages of logratio analysis. A
deeper discussion of this topic falls outside the scope of this paper, but the interested reader is referred to Aitchison (1986)
and Pawlowsky-Glahn et al. (2015) for further information. The Euclidean distance between logratios is also known as the
'Aitchison distance'. Discordance can be redefined as the Aitchison distance from the measured logratios to the concordia
line. We here introduce two ways to do so. A first option is to simply measure the distance along a perpendicular line to the
concordia curve (Figure 6):

$$d_a = dx(t_{68}) \sin\left(\arctan\left[\frac{dy(t_{76})}{dx(t_{68})}\right]\right) \tag{6}$$

where

$$dx(t) = \log\left[\frac{^{206}Pb}{^{238}U}\right] - \log[\exp(\lambda_{238}t) - 1]$$
$$\text{and } dy(t) = \log\left[\frac{^{207}Pb}{^{206}Pb}\right] - \log\left[\frac{^{235}U}{^{238}U}\frac{\exp(\lambda_{235}t) - 1}{\exp(\lambda_{238}t) - 1}\right] \tag{7}$$





This definition produces a parallel band around the concordia line in logarithmic Tera-Wasserburg space. In contrast with $d_r$, $d_t$, $d_{sk}$, the $d_a$ criterion is less strict at both the young and old extremes of the geological timescale, and more strict during the Proterozoic Era (Figure 3), when the U–Pb method is most reliable.

The perpendicular Aitchison distance criterion does not take into account the analytical precision of the isotopic measurements. To address this issue, we can also measure the Aitchison distance along a line connecting the measured logratio and the maximum likelihood composition on the concordia line:

$$d_c = \text{sgn}[t_{76} - t_{68}]\sqrt{dx(t_c)^2 + dy(t_c)^2} \qquad (8)$$

where sgn[∗] stands for the "sign of ∗", which produces positive values for measurements that plot above the concordia line,
and negative values for measurements that plot below it.

## 5   Application to a compilation of detrital zircon U–Pb data

It is difficult to ascertain the mechanism causing discordance in any particular zircon grain. Therefore, it is unclear which of the definitions in Sections 3 and 4 is 'correct'. All we can is do is apply the methods to real samples and investigate their outcomes. This Section will apply the six discordance filters to a compilation of 10,000 detrital zircon U–Pb analyses that were acquired
at the London Geochronology Centre (LGC) at University College London.

The data come from field areas on all seven continents. They were acquired by LA-ICP-MS, on either an Agilent 7700x (early samples) or an Agilent 7900 (recent samples) instrument, which was coupled to a New Wave NWR193 excimer laser. Analytical conditions varied slightly between different samples but generally used a 25 or 35 micron laser spot, Plešovice zircon as a primary standard (Sláma et al., 2008) and, in most cases, GJ-1 zircon as a secondary standard (Jackson et al., 2004).
Data reduction was done with GLITTER (Griffin et al., 2008). The data were not subjected to any common Pb correction or other filters, apart from a visual inspection to remove the most extreme outliers ($< 1\%$ of the data).

Figure 7 shows the frequency distribution of the complete, unfiltered dataset as a kernel density estimate. The $^{207}$Pb/$^{235}$U, $^{206}$Pb/$^{238}$U and concordia age spectra all look similar. However, the $^{207}$Pb/$^{206}$Pb age distribution deviates from the other three chronometers. It reduces the prominence of the young age components, and inflates the old end of the age spectrum.
Figure 8 applies the six discordance filters to this database. In order to emphasise the difference between the six discordance definitions whilst treating them on an equal footing, each of the filters was adjusted until half of the data were removed. This was achieved by discordance cutoffs of $d_t$ = 51 Myr, $d_r$ = 0.0618, $d_p$ = 0.084, $d_{sk}$ = 0.00225, $d_a$ = 0.0219 and $d_c$ = 0.0297.

There are noticeable differences between the density estimates. As expected from the theoretical considerations laid out in Sections 3 and 4, the relative age filter greatly suppresses the younger age components ($< 1.5$ Ga) relative to the older parts
of the age spectrum ($> 1.5$ Ga). The Stacey and Kramers (1975) filter has the opposite effect. It suppresses the Archaean age component by $\sim$20% whilst further increasing the prominence of the Neoproterozoic and Phanerozoic modes.

The discordance definitions based on the absolute age difference and logratio distances have a comparatively minor effect on the shape of the age spectrum. All six discordance filters reduce the difference between the $^{207}$Pb/$^{206}$Pb and concordia age spectra except for the p-value filter, which actually exacerbates the problem.




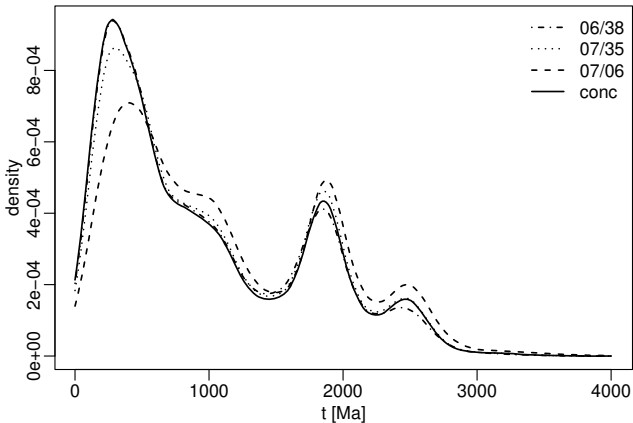

**Figure 7.** Four superimposed kernel density estimates (with 60 Ma bandwidth) for 10,000 unfiltered detrital zircon U–Pb dates. The $^{207}$Pb/$^{235}$U, $^{206}$Pb/$^{238}$U and concordia age spectra are similar. However the KDE of the $^{207}$Pb/$^{206}$Pb data stands apart from the other three curves. It deviates both at the young end of the age spectrum (which it suppresses), and at the old end (which it inflates).

If the degree of similarity between the four chronometers is taken as a measure of their success, then the concordia distance filter ($d_c$) is the most successful discordance criterion. Apart from a minor difference at the youngest end of the age spectrum, the four different chronometers yield virtually identical age distributions after passing through this filter.

Figure 8 removed 50% of the data, in order to emphasise the differences between the six filters. In real applications, less stringents discordance filters are usually applied. As mentioned in the introduction, most current detrital zircon studies apply a 10%–30% relative age cutoff. Using the test data, we can evaluate the equivalent values for the $d_t$, $d_t$, $d_{sk}$, $d_a$ and $d_c$ criteria (Table 1). For example, a relative age filter of 10% removes the same fraction of the test data as an absolute age filter with $d_t = 83.5$ Ma, a Stacey-Kramers filter with $d_{sk} = 0.381\%$, a perpendicular Aitchison filter with $d_a = 3.61\%$, or a concordia distance filter with $d_c = 3.67\%$.

The p-value discordance filter has been omitted from this comparison for two reasons. First, the use of this filter is discouraged for reasons that have been discussed before. Second, the p-value cutoffs that are equivalent to any given relative age difference are highly laboratory dependent, with precise equipment requiring different $d_p$-cutoffs than imprecise instruments. The other five discordance filters are more universally applicable. So using a different set of test data should only make a relatively minor difference to the values in Table 1.

## 6 Conclusions

This paper compared four U–Pb clocks and six discordance filters.

1. The $^{206}$Pb/$^{238}$U clock is most precise at the young end of the geologic timescale.

2. The $^{207}$Pb/$^{206}$Pb method is more precise than the $^{206}$Pb/$^{238}$U method before the Neoproterozoic.



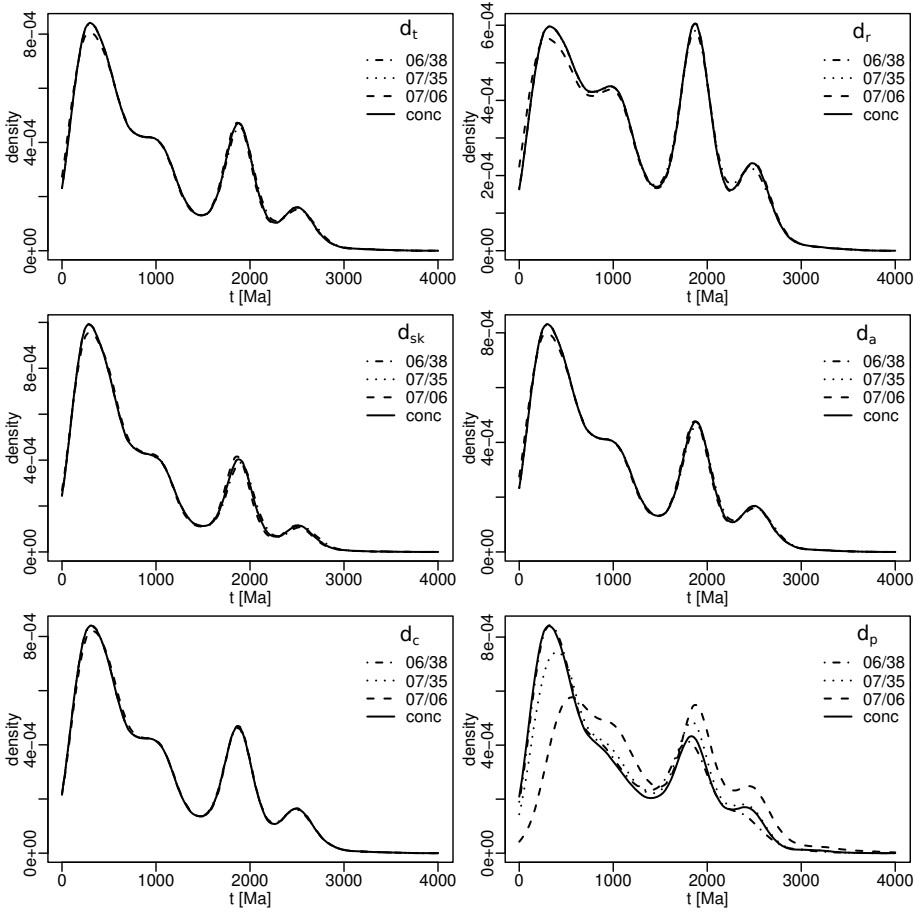

**Figure 8.** Filtered U–Pb age spectra for the test data, removing the 50% most discordant grains according to the six discordance filters reviewed in this paper. Note how the concordia distance filter ($d_c$, fifth panel) exhibits the smallest difference between the four dating methods (including $^{207}$Pb/$^{206}$Pb), whereas the p-value filter ($d_p$, sixth panel) generates the greatest differences between them. This observation suggests that the $d_c$ criterion is the best and the $d_p$ criterion the worst of the six methods. Also note how the relative age filter ($d_r$) strongly inflates the older half of the age spectrum at the expense of the younger half. The Stacey-Kramers filter ($d_{sk}$) has the opposite effect, as expected from Figure 3.

3. The $^{207}$Pb/$^{235}$U clock offers no advantage over the other two methods.

4. The single grain concordia age is applicable to the entire span of geologic time and always offers the best precision. It approaches the $^{206}$Pb/$^{238}$U age as time approaches zero, and the $^{207}$Pb/$^{206}$Pb age as time approaches infinity.

The six discordance filters include three existing ones and three new ones.

1. The relative age discordance $d_r$ is the most widely used criterion today. It is more likely to remove young grains than old ones.





| $d_r$ | $d_t$ | $d_{sk}$ | $d_a$ | $d_c$ |
|---|---|---|---|---|
| -10 | -60.4 | -0.287 | -2.59 | -2.62 |
| -5 | -38.8 | -0.175 | -1.67 | -1.69 |
| -4 | -32.3 | -0.143 | -1.39 | -1.41 |
| -3 | -25 | -0.108 | -1.08 | -1.09 |
| -2 | -17.3 | -0.0681 | -0.747 | -0.758 |
| -1 | -8.33 | -0.0328 | -0.356 | -0.361 |
| 0 | 0 | 0 | 0 | 0 |
| 1 | 9.31 | 0.0367 | 0.395 | 0.406 |
| 2 | 18.5 | 0.0773 | 0.795 | 0.816 |
| 3 | 28.1 | 0.117 | 1.2 | 1.23 |
| 4 | 36.7 | 0.151 | 1.58 | 1.6 |
| 5 | 43.9 | 0.189 | 1.9 | 1.94 |
| 10 | 83.5 | 0.381 | 3.61 | 3.67 |
| 15 | 117 | 0.538 | 5.02 | 5.16 |
| 20 | 151 | 0.736 | 6.54 | 6.7 |
| 25 | 187 | 0.949 | 8.01 | 8.24 |
| 30 | 230 | 1.18 | 9.93 | 10.2 |
| 40 | 328 | 1.74 | 14.2 | 14.8 |
| 50 | 463 | 2.47 | 20.3 | 21.6 |

**Table 1.** Conversion table for the different discordance filters, constructed using the test data. All discordance values are expressed as %, except for $d_t$, which is expressed in Ma. This table allows the reader to select a discordance cutoff that removes the same fraction of their data as the relative age cutoff ($d_r$) that they may have applied in the past.

2. The absolute age discordance $d_t$ is not widely used. But it illustrates the dramatic effect that the discordance definition can have on the filtered age distibutions. Compared with the relative age filter, it is more likely to reject old grains, and less likely to reject young ones.

3. The p-value based discordance filter $d_p$ may have intuitive appeal as an 'objective' definition. But it has an undesirable negative effect on the precision and accuracy of the filtered results.

4. The Stacey-Kramers discordance filter $d_{sk}$ assumes that discordance is solely caused by common Pb contamination. If this assumption is correct, then the $d_{sk}$ filter will produce the most accurate age distributions, provided that a Stacey and Kramers (1975) common Pb correction is applied to the filtered data.

5. The perpendicular Aitchison distance $d_a$ is a useful vehicle to illustrate the application of logratio statistics to detrital zircon U–Pb geochronology. It produces a parallel acceptance zone around the (log-transformed) concordia line. This filter is most likely to reject 'middle aged' zircon grains, between 1000 and 2000 Ma, where the age resolving power of

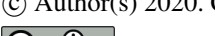



the U–Pb method is greatest. Above and below this interval, the $d_a$ criterion is more forgiving. This behaviour is desirable because natural samples tend to exhibit more age discordance below 1000 Ma and above 2000 Ma than between these dates.

6. The concordia distance $d_c$ is a modified version of the $d_a$ criterion that takes into account the (correlated) uncertainties of the U–Pb isotopic composition. Its effects on the U–Pb age distributions are more difficult to visualise but are similar

to those of the $d_a$ criterion. Applying the $d_c$ filter to the test data shows that it minimises the difference between the $^{207}$Pb/$^{235}$U, $^{206}$Pb/$^{238}$U, $^{207}$Pb/$^{206}$Pb and concordia age spectra. Therefore the $d_c$ criterion simplifies the interpretation of detrital zircon U–Pb age spectra.

All the discordance filters presented in this paper (both old and new) have been implemented in `IsoplotR` (Vermeesch, 2018), a geochronological toolbox written in the `R` language. Further details about this implementation are provided in Ap-

pendix B.

*Code and data availability.* `IsoplotR` is free software released under the GPL-3 license. The package and its source code are available from `https://cran.r-project.org/package=IsoplotR`. The test data can be downloaded from `https://github.com/pvermees/discpaper/`.

## Appendix A:  Comparing the precision of the $^{207}$Pb/$^{235}$U, $^{206}$Pb/$^{238}$U, $^{207}$Pb/$^{206}$Pb and concordia clocks

The uncertainty of a U–Pb date depends on three factors:

1. the age and, hence, the true isotopic ratio;

2. the sensitivity of the ion detectors to U and Pb; and

3. the dwell times used to measure the different isotopes.

These three factors vary between samples, and between labs. In order to explore their effects, let us first define the following

parameters:

– $t_{68}$, $t_{75}$ and $t_{76}$: the $^{206}$Pb/$^{238}$U, $^{207}$Pb/$^{235}$U and $^{207}$Pb/$^{206}$Pb ages (in Ma);

– $\lambda_{38}$ and $\lambda_{35}$: the decay constants of $^{238}$U and $^{235}$U (in Ma$^{-1}$);

– $R_{85}$: the natural $^{238}$U/$^{235}$U ratio;

– $R_{68}$, $R_{75}$ and $R_{76}$: the true $^{206}$Pb/$^{238}$U, $^{207}$Pb/$^{235}$U and $^{207}$Pb/$^{206}$Pb atomic ratios;

– $r_{68}$, $r_{75}$ and $r_{76}$: the measured $^{206}$Pb/$^{238}$U, $^{207}$Pb/$^{235}$U and $^{207}$Pb/$^{206}$Pb signal ratios;



- $f_U^{Pb}$: the fractionation factor between Pb and U;

- $d_{38}^{06}$: the dwell time ratio of $^{206}$Pb and $^{238}$U;

- $d_{06}^{07}$: the dwell time ratio of $^{207}$Pb and $^{206}$Pb;

- $n_{06}$, $n_{07}$ and $n_{38}$: the number of $^{206}$Pb, $^{207}$Pb and $^{238}$U ions counted during a measurement.

Then the true isotope ratios are given by:

$$R_{68} = \exp(\lambda_{38} t_{68}) - 1 \tag{A1}$$

$$R_{75} = \exp(\lambda_{35} t_{75}) - 1 \tag{A2}$$

$$R_{76} = \frac{1}{R_{85}} \frac{R_{75}}{R_{68}} \tag{A3}$$

and the measured ratios by:

$$r_{68} = d_{38}^{06} f_U^{Pb} R_{68} \tag{A4}$$

$$r_{75} = d_{06}^{07} d_{38}^{06} f_U^{Pb} R_{75} \tag{A5}$$


$$r_{76} = d_{06}^{07} R_{76} \tag{A6}$$

so that the predicted $^{206}$Pb and $^{207}$Pb ion counts can be written as:

$$n_{06} = n_{38} d_{38}^{06} f_U^{Pb} R_{68} \tag{A7}$$

$$n_{07} = n_{06} d_{06}^{07} R_{76} \tag{A8}$$

Assuming that all the ions are measured by Secondary Electron Multiplier (SEM), with analytical uncertainties that are governed by Poissonian shot noise:

$$\left( \frac{\sigma[r_{68}]}{r_{68}} \right)^2 = \frac{1}{n_{38}} + \frac{1}{n_{06}} \tag{A9}$$





$$\left(\frac{\sigma[r_{75}]}{r_{75}}\right)^2 = \frac{1}{n_{38}} + \frac{1}{n_{07}} \tag{A10}$$

$$\left(\frac{\sigma[r_{76}]}{r_{76}}\right)^2 = \frac{1}{n_{06}} + \frac{1}{n_{07}} \tag{A11}$$

then the standard errors of the signal ratios ratios are given by:

$$\sigma[r_{68}] = \frac{n_{06}}{n_{38}} \sqrt{\frac{1}{n_{38}} + \frac{1}{n_{06}}} \tag{A12}$$


$$\sigma[r_{75}] = R_{85} \frac{n_{07}}{n_{38}} \sqrt{\frac{1}{n_{38}} + \frac{1}{n_{07}}} \tag{A13}$$

$$\sigma[r_{76}] = \frac{n_{07}}{n_{06}} \sqrt{\frac{1}{n_{06}} + \frac{1}{n_{07}}} \tag{A14}$$

Finally, the uncertainties of the age estimates are given by standard error propagation:

$$\sigma[t_{68}] = \frac{\partial t_{68}}{\partial r_{68}} \sigma[r_{68}] \tag{A15}$$

$$\sigma[t_{75}] = \frac{\partial t_{75}}{\partial r_{75}} \sigma[r_{75}] \tag{A16}$$

$$\sigma[t_{76}] = \frac{\partial t_{76}}{\partial r_{76}} \sigma[r_{76}] \tag{A17}$$

where

$$\frac{\partial t_{68}}{\partial r_{68}} = \frac{1}{\lambda_{38}(1 + R_{68})} \frac{1}{d_{38}^{06} f_U^{Pb}} \tag{A18}$$

$$\frac{\partial t_{75}}{\partial r_{75}} = \frac{1}{\lambda_{35}(1 + R_{75})} \frac{1}{d_{06}^{07} f_U^{Pb}} \tag{A19}$$

$$\frac{\partial t_{76}}{\partial r_{76}} = \frac{R_{85} R_{68}^2}{(\partial R_{75}/\partial t_{75})R_{68} - R_{75}(\partial R_{68}/\partial t_{68})} \frac{1}{d_{06}^{07}} \tag{A20}$$

Figure 2 shows the result of these calculations using realistic values of $n_{38}$, $f_U^{Pb}$ and $d_{06}^{07}$, which yield an outcome that is similar to the test data, and to the empirical results of Zimmermann et al. (2018).




## Appendix B: Implementation in `IsoplotR`

`IsoplotR` can be accessed either from the command line, or via a graphical user interface (GUI), either offline or online
(`http://isoplotr.london-geochron.com`). The discordance filters are accessible via both methods. In the GUI, the discordance can be tabulated via the `age` function, and has also been incorporated in `IsoplotR`'s other functions, including its concordia and weighted mean calculation algorithms. Further details about these options are provided under the 'Options' menu. To access the same functionality from the command line, we must first install `IsoplotR` from the Comprehensive R Archive Network (CRAN):

```
install.packages('IsoplotR')
```

Once installed, we need to add the package to our working environment:

```
library(IsoplotR)
```

Loading the test data into memory:

```
UPb <- read.data('data.csv',method='U-Pb',format=2)
```

Now we can calculate the discordance using `IsoplotR`'s `discfilter` function. For example, to compute the relative age discordance ($d_r$):

```
tr <- age(UPb,discordance=discfilter(option='r'))
```

which produces a $10000 \times 9$ table whose first eight columns list the $^{207}$Pb/$^{235}$U, $^{206}$Pb/$^{238}$U, $^{207}$Pb/$^{206}$Pb and concordia ages and their uncertainties, and whose ninth column lists the relative age discordance as percentages. Similarly, to compute the
concordia distance ($d_c$):

```
tc <- age(UPb,discordance=discfilter(option='c'))
```

Plotting a KDE of the single grain concordia ages that pass the perpendicular Aitchison filter with $-1.67 \leq d_a \leq 5.02$:

```
df <- discfilter(option='c',cutoff=c(-1.69,5.16))
kde(UPb,type=5,cutoff.disc=df)
```

Apply a Stacey-Kramers common Pb-correction to the data after applying a Stacey-Kramers discordance filter with $0.285 \leq d_{sk} \leq 1.16$:

```
df <- discfilter(option='sk',cutoff=c(-0.285,1.16))
kde(UPb,common.Pb=3,cutoff.disc=df)
```

If the dataset includes $^{204}$Pb (which is not the case for the test data), then we can also apply a discordance filter *after* the
common Pb correction. For example:




```
df <- discfilter(option='r',before=FALSE,cutoff=c(-5,15))
kde(UPb,common.Pb=3,type=4,cutoff.76=1200,cutoff.disc=df)
```

where `option='r'` triggers the relative age filter ($d_r$), `common.Pb=3` applies a Stacey-Kramers type common Pb correc-
tion, `type=4` uses the $^{206}$Pb/$^{238}$U-age for young grains and the $^{207}$Pb/$^{206}$Pb-age for old ones, and `cutoff.76` marks the age
(in Ma) at which to switch from the $^{206}$Pb/$^{238}$U to the $^{207}$Pb/$^{206}$Pb method. Further information about these functions can be
obtained from the built-in documentation:

```
?IsoplotR
?discfilter
?kde
```

Note that the examples shown here may take a few minutes to complete due to the large size of the test dataset.

*Author contributions.*   This is a single author publication.

*Competing interests.*   The author has no competing interests.

*Acknowledgements.*   The writing of this paper was triggered by a stimulating email conversation with Chris Spencer and Steve Puetz. This
research was supported by NERC standard grant #NE/T001518/1 ('Beyond Isoplot').





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
