# Peer review of "Redefining U-Pb discordance"

_Geochronology, 2020_

## Referee Comment (RC1) · Ping Wang (Referee) · 15 Jan 2021

The manuscript is potentially an important scientific contribution that will be of wide interest to broad geological audience. Despite the increasingly widespread use of the detrital zircon U-Pb geochronology for provenance signature, there are some ambiguities that may have strong effects on data interpretation, such as "discordance". The author presents six definitions of the discordance and make a novel comparison using age spectrum of real samples. The result seems clear and provide useful implications for future application. I suggest the manuscript should be accepted after minor revision.

Some comments:

Line 35: I think the superscript "204" is not necessary.

[Figure]

Line 94: It requires another equation for the concordia age with 6/8 and 7/6 ratios.

Line 101: It will be much clearer if you can list the six definitions for different discordance filters in a table.

Figure 2: Is the concorida age (tc) calculated by 6/8 and 7/6 ratios or 6/8 and 7/5 ratios?

Figure 3: The top figure is good! But it will be better if you can give an envelope (above and below the concordia line) for each discordance filter because the discordance can be negative in nature. In contrast, the bottom figure is confusing in its present form. The absolute age filter (dt) may not be the best filter but it looks like a robust one in the figure.

Figure 5: Are 0.84, 0.85 and 0.87 the intersections for the three lines? What do the arrows mean?

Figure 6: The axis values may not be in logarithmic space. The log (7/6) should be negative value as shown in Figure 3. It's better to add the label dx and dy for the dash lines.

Equation 7: exp(lambda235t)-1 and exp(lambda238t)-1 should be in parentheses.

---

## Referee Comment (RC2) · Keith Sircombe (Referee) · 31 Jan 2021

— General comments —

Detrital zircon geochronology (DZG) analysis is a very active field of research and papers like this one are very important to continually improve the fundamental mathematic basis of the research regardless of analytical method. The review and proposed solutions presented by this paper will make a big step toward an ideal state where acquired DZG data are treated with a consistent methodology and resulting interpretations are grounded in a thorough understanding of the underlying assumptions and potential biases.

The paper provides a suitably brief overview of the current approaches to treating ac-

quired DZG isotopic data and a presentation of a proposed 'best practice' solution for optimising data and avoiding arbitrary and subjective thresholds. Given the high levels of research activity using DZG analysis, this paper will find a broad audience of interest both from analysts acquiring data and scientists applying the results to geological problems.

I recommend publication with minor revisions.

— Specific comments —

* Although there are a couple of mentions (line 170, Table 1), there isn't much detailed discussion about the treatment of reversely or negative discordant data. It would be useful to know, even if only for the sake of completeness, if the various discordance filters cause any biasing of negative discordance relative to positive discordance. (Or another aspect: the absolute discordance values in Table 1 are asymmetrical for negative and positive values – does this have any impact on how the data are treated?). I recognise this may be a whole topic into itself, but some advice in the paper would be useful.

* The recommendation to use the Concordia distance is based on the argument that it appears to minimise 'the difference between the $^{207}Pb/^{235}U$, $^{206}Pb/^{238}U$, $^{207}Pb/^{206}Pb$ and concordia age spectra' (Line 235) via a visual assessment of Figure 8. The corollary is that the other discordance filters illustrated in Figure 8 have a greater difference between spectra – again via a visual assessment of the figure that may not be especially clear on screen. It would be great to quantify these differences between discordance filters – perhaps a Kolmogorov–Smirnov test, or similar, would provide an objective measure of 'best practice' among these discordance filters.

* It would also be interesting to see the discordance filters being further 'stress tested' with other datasets especially in the < 1000 Ma and >3000 Ma age ranges. How well does the concordia distance filter work when there is a wide range of Neoproterozoic and Phanerozoic grains present? For instance, is there any discernible biasing

between ∼1000 Ma and ∼100 Ma age groups that may impact interpretations for a detrital zircon study focusing on this age range?

— Technical comments —

Just a few minor notes, and largely an individual style choice, so no big deal:

* Line 24-26. The term 'chronometers' is introduced here after using 'decay systems' and then used again to describe the U-Pb systems in Line 73. Perhaps a tidier and consistent use of terminology might help those unfamiliar with the geochronology methodology.

* Line 30: should be singular '. . .fulfil this requirement. . .'

* Line 30: The intent of the last sentence might be clearer if written 'Those that fail to meet this requirement are 'discordant'.'

* Line 120: I think I get what you are aiming for in this part, but the term 'unfair' strikes me as the incorrect phrase. 'Fairness' is a subjective concept more pertinent to ethical or legal considerations. Perhaps this part needs to be better phrased to reflect that a method favouring an imprecise method over a precise method is simply poor science that will encourage biasing – I'm sure there are better and more precise, ways of putting it, but the words escape me at the moment. . . Perhaps it something related to 'small-study effects' where small studies, often earlier ones with less precision, can report (or be interpreted with) larger effects.

* Line 157: this might be better phrased as 'We introduce two ways to do so here.' And while the use of 'we' in the manuscript is a good active narrative tool, in this particular sentence it may be better as 'I' because you are the sole author introducing an idea to the readers.

---

## Author Comment (AC1) · 11 Feb 2021

I would like to thank the reviewer for his positive review and his sharp eyes, which spotted a few errors that had escaped my attention. I will follow all his recommendations in the revised manuscript.

Line 35: I think the superscript "204" is not necessary

$^{204}$Pb can be used in discordia regression. But to avoid confusion, I will change "$^{238}$U–$^{204,6,7}$Pb space" to "U–Pb isotope space".

Line 94: It requires another equation for the concordia age with 6/8 and 7/6

ratios.

The concordia age calculation can either be done in Wetherill space or in Tera Wasserburg space. The Wetherill version is simpler but I will follow the reviewer's suggestion and replace Equation 1 with the Tera-Wasserburg formulation because the paper contains several Tera-Wasserburg concordia diagrams but no Wetherill diagram.

$$S = \begin{bmatrix} r_{86} - 1/R_{68}(t_c) \\ r_{76} - R_{58}R_{75}(t_c)/R_{68}(t_c) \end{bmatrix}^T \begin{bmatrix} \sigma[r_{86}]^2 & \sigma[r_{86}, r_{76}] \\ \sigma[r_{86}, r_{76}] & \sigma[r_{76}]^2 \end{bmatrix}^{-1} \begin{bmatrix} r_{86} - 1/R_{68}(t_c) \\ r_{76} - R_{58}R_{75}(t_c)/R_{68}(t_c) \end{bmatrix}$$

Line 101: It will be much clearer if you can list the six definitions for different discordance filters in a table.

I will add the requested table to the conclusions section:

| definition | description | comment |
|---|---|---|
| $d_r = 1 - t_{68}/t_{76}$ | relative age difference | biases against young samples |
| $d_{sk} = 1 - r_{86}/r_{86}^*$ | fraction of common Pb | biases against old samples |
| $d_p = \mathsf{Prob}\left(s > S \vert S \sim \chi_2^2\right)$ | p-value of concordance | biases against precise measurements |
| $d_t = t_{76} - t_{68}$ | absolute age difference | allows negative ages |
| $d_a = dx(t_{68}) \sin\left(\arctan\left[\frac{dy(t_{76})}{dx(t_{68})}\right]\right)$ | Aitchison distance | most strict for 'middle aged' samples |
| $d_c = \mathsf{sgn}[t_{76} - t_{68}]\sqrt{dx(t_c)^2 + dy(t_c)^2}$ | concordia distance | least biased |

Figure 2: Is the concordia age (tc) calculated by 6/8 and 7/6 ratios or 6/8 and 7/5 ratios?

See my response to the comment about line 94. $t_c$ can be calculated using either set of ratios.

Figure 3: The top figure is good! But it will be better if you can give an envelope (above and below the concordia line) for each discordance filter because the discordance can be negative in nature. In contrast, the bottom figure is confusing in its present form. The absolute age filter (dt) may not be the best filter but it looks like a robust one in the figure.

I will follow the reviewer's suggestion and extend the envelope below the concordia line (see Figure 1). I will also remove the bottom panel. The absolute age filter may look unreasonable, but actually outperforms the relative age filter!

Figure 5: Are 0.84, 0.85 and 0.87 the intersections for the three lines? What do the arrows mean?

I have added an inset and removed the arrows to clarify the figure (Figure 2).

Figure 6: The axis values may not be in logarithmic space. The log (7/6) should be negative value as shown in Figure 3. It's better to add the label dx and dy for the dash lines.

Done (Figure 3).

Equation 7: $\exp(\lambda_{235}t) - 1$ and $\exp(\lambda 238t) - 1$ should be in parentheses.

The revised manuscript will introduce further shorthand notation to shorten the equations and remove the number of nested brackets:

$$d_a = dx(t_{68}) \sin\left(\arctan\left[\frac{dy(t_{76})}{dx(t_{68})}\right]\right)$$

where

$$dx(t) = \ln[r_{86}] + \ln[R_{68}(t)] \text{ and } dy(t) = \ln[r_{76}] - \ln\left[R_{58}\frac{R_{75}(t)}{R_{68}(t)}\right]$$

with $R_{68}(t) = \exp(\lambda_{238}t) - 1$, $R_{75}(t) = \exp(\lambda_{235}t) - 1$, and $R_{58} = {}^{235}\text{U}/{}^{238}\text{U}$.

[Figure]

**Fig. 1.**

[Figure]

**Fig. 2.**

Labels on the figure: $t_{76}$, $dy(t_{76})$, $d_a$, $d_c$, $t_c$, $t_{68}$, $dx(t_{68})$.

**Fig. 3.**

---

## Author Comment (AC2) · 11 Feb 2021

I agree with all the reviewer's comments and will address them in the revised manuscript as follows:

* Although there are a couple of mentions (line 170, Table 1), there isn't much detailed discussion about the treatment of reversely or negative discordant data. It would be useful to know, even if only for the sake of completeness, if the various discordance filters cause any biasing of negative discordance relative to positive discordance. (Or another aspect: the absolute discordance values in Table 1 are asymmetrical for negative and positive values – does this have any impact on how the data are treated?). I

[Figure]

recognise this may be a whole topic into itself, but some advice in the paper would be useful.

I will add the following information about negative discordance:

1. I will extended the discordance envelope of Figure 3 to negative values, in response to a request by Reviewer Wang (see Figure 1 of this response).

2. I will added a paragraph to the discussion of the Stacey-Kramers discordance definition, pointing out that, even though this definition is mathematically able to produce negative discordance values, such values cease to have a geologically meaningful interpretation, because it is impossible for minerals to inherit negative amounts of common Pb. Thus it may be advisable to set a minimum cutoff of $d_{sk} > 0$ when using the Stacey-Kramers filter.

3. If will point out that both the Stacey-Kramers and absolute age filter may let physically impossible negative $^{207}$Pb/$^{206}$Pb ages pass through them. This may be seen as an argument against these filters.

   * The recommendation to use the Concordia distance is based on the argument that it appears to minimise 'the difference between the $^{207}$Pb/$^{235}$U, $^{206}$Pb/$^{238}$U, $^{207}$Pb/$^{206}$Pb and concordia age spectra' (Line 235) via a visual assessment of Figure 8. The corollary is that the other discordance filters illustrated in Figure 8 have a greater difference between spectra – again via a visual assessment of the figure that may not be especially clear on screen. It would be great to quantify these differences between discordance filters – perhaps a Kolmogorov-Smirnov test, or similar, would provide an objective measure of 'best practice' among these discordance filters.

[Figure]

This is an excellent suggestion. I will add quantile-quantile plots and Kolmogorov-Smirnov statistics to Figure 8, not to inter-compare the different chronometers, but rather to compare each of the filtered datasets against the unfiltered dataset. This exercise provides a much stronger argument for the concordia distance than the previous version of Figure 8 did. I have attached the revised figure to my response (Figure 2 below).

> \* It would also be interesting to see the discordance filters being further 'stress tested' with other datasets especially in the $< 1000$ Ma and $> 3000$ Ma age ranges. How well does the concordia distance filter work when there is a wide range of Neoproterozoic and Phanerozoic grains present? For instance, is there any discernible biasing between $\sim$1000 Ma and $\sim$100 Ma age groups that may impact interpretations for a detrital zircon study focusing on this age range?

I have contacted Reviewer Sircombe and his colleagues at Geoscience Australia to follow up on this comment. They have kindly provided me with a 1600 sample, 70000 grain SIMS dataset of superior quality to the 10000 grain LAICPMS dataset used in the previous version of the paper. The new database contains more grains at the young and old end of the age spectrum, confirming consistent behaviour across all time scales, i.e. the relative age discordance filter keeps rejecting a progressively large proportion of the young grains, and the Stacey-Kramers filter does the opposite (see Figure 2 of this response).

I will also extend the time scale of Figure 3 from 100 to 10 Myr, providing an extended view into the young timescales. This highlights a problem with the absolute age and Stacey-Kramers filters, which allow physially impossible negative ages to pass through (see Figure 1 of this response).

> \* Line 24-26. The term 'chronometers' is introduced here after using 'decay systems' and then used again to describe the U-Pb systems in Line 73. Perhaps a tidier and consistent use of terminology might help those unfamiliar with the geochronology methodology.

I have used hyphens (–) whenever decay systems are referred to and ratios to refer to the chronometers. For example, the $^{235}$U–$^{207}$Pb decay system forms the basis of the $^{207}$Pb/$^{235}$U chronometer.

Reviewer Sircombe's remaining comments are language suggestions, which will all be incorporated into the revised paper.

[Figure]

**Fig. 1.**

**Fig. 2.**

density

$d_r$   KS=0.218

$d_t$   KS=0.0556

$d_{sk}$   KS=0.106

$d_a$   KS=0.0693

$d_c$   KS=0.046

t [Ma]

quantiles (unfiltered) [Ma]

quantiles (filtered) [Ma]

---

## Author Response (AR1)

Prof. Pieter Vermeesch
University College London
+44 (0)20 3108 6369
http://ucl.ac.uk/~ucfbpve/

15 February 2021

Dr. Michael Dietze
Associate Editor
*Geochronology*

Dear Dr. Dietze,

I hereby submit the revised version of gchron-2020-38 to *Geochronology*. In addition to the changes that are mentioned in my responses to the two reviewers, I have made the following amendments:

1. The title was changed to "Redefining U–Pb discordance", which draws attention to the most important objective of the paper.

2. A new figure was added to the appendix, showing `IsoplotR`'s GUI. This should remove any concerns that the new discordance filters are more difficult to use than the old ones.

I hope that you find the new version acceptable for publication in *Geochronology*

Please let me know if you have any other comments or requests,

Sincerely yours,

Pieter Vermeesch

---

## Author Response (AR2)

Prof. Pieter Vermeesch
University College London
+44 (0)20 3108 6369
http://ucl.ac.uk/~ucfbpve/

19 February 2021

Dr. Michael Dietze
Associate Editor
*Geochronology*

Dear Dr. Dietze,

I hereby submit the final version of gchron-2020-38 to *Geochronology*. As discussed by email, I have reverted to the original title "On the treatment of discordant detrital zircon U–Pb data".

Sincerely yours,

Pieter Vermeesch